# Antibacterial activities of Miang extracts against selected pathogens and the potential of the tannin-free extracts in the growth inhibition of *Streptococcus mutans*

Aliyu Dantani Abdullahi[1], Kridsada Unban[2], Chalermpong Saenjum[3], Pratthana Kodchasee[4], Napapan Kangwan [5], Hathairat Thananchai[6], Kalidas Shetty[7], Chartchai Khanongnuch [4,8,9]*

1 Interdisciplinary Program in Biotechnology, The Graduate School, Chiang Mai University, Muang, Chiang Mai, Thailand, 2 Faculty of Agro-Industry, Division of Food Science and Technology, School of Agro-Industry, Chiang Mai University, Muang, Chiang Mai, Thailand, 3 Faculty of Pharmacy, Department of Pharmaceutical Sciences, Chiang Mai University, Muang, Chiang Mai, Thailand, 4 Research Center for Multidisciplinary Approaches to Miang, Multidisciplinary Research Institute (MDRI), Chiang Mai University, Muang, Chiang Mai, Thailand, 5 Division of Physiology, School of Medical Sciences, University of Phayao, Phayao, Thailand, 6 Faculty of Medicine, Department of Microbiology, Chiang Mai University, Muang, Chiang Mai, Thailand, 7 Faculty of Agriculture, Department of Plant Sciences, North Dakota State University, Fargo, North Dakota, United States of America, 8 Faculty of Science, Department of Biology, Chiang Mai University, Chiang Mai, Thailand, 9 Research Center of Microbial Diversity and Sustainable Utilization, Chiang Mai University, Chiang Mai, Thailand

* chartchai.k@cmu.ac.th, ck_biot@yahoo.com

## Abstract

Bacterial pathogens have remained a major public health concern for several decades. This study investigated the antibacterial activities of Miang extracts (at non-neutral and neutral pH) against *Bacillus cereus* TISTR 747, *Escherichia coli* ATCC 22595, *Salmonella enterica* serovar Typhimurium TISTR 292 and *Streptococcus mutans* DMST 18777. The potential of Polyvinylpolypyrrolidone (PVPP)-precipitated tannin-free Miang extracts in growth-inhibition of the cariogenic *Streptococcus mutans* DMST 18777 and its biofilms was also evaluated. The tannin-rich fermented extracts had the best bacterial growth inhibition against *S. mutans* DMST 18777 with an MIC of 0.29 and 0.72 mg/mL for nonfilamentous fungi (NFP) Miang and filamentous-fungi-processed (FFP) Miang respectively. This observed anti-streptococcal activity still remained after PVPP-mediated precipitation of bioactive tannins especially, in NFP and FFP Miang. Characterization of the PVPP-treated extracts using High performance liquid chromatography quadrupole-time of flight-mass spectrometry (HPLC-QToF-MS) analysis, also offered an insight into probable compound classes responsible for the activities. In addition, Crystal violet-staining also showed better $IC_{50}$ values for NFP Miang (4.30 ± 0.66 mg/mL) and FFP Miang (12.73 ± 0.11 mg/mL) against *S. mutans* DMST 18777 biofilms *in vitro*. Homology modeling and molecular docking analysis using HPLC-MS identified ligands in tannin-free Miang supernatants, was performed against modelled *S. mutans* DMST 18777 sortase A enzyme. The *in silico* analysis suggested that the inhibition by NFP and FFP Miang might be attributed to the presence of ellagic acid, flavonoid

**Data Availability Statement:** All relevant data are within the paper and its Supporting information files.

**Funding:** The author(s) received no specific funding for this work.

**Competing interests:** The authors have declared that no competing interests exist.

aglycones, and glycosides. Thus, these Miang extracts could be optimized and explored as natural active pharmaceutical ingredients (NAPIs) for applications in oral hygienic products.

## Introduction

The United Nations' drive to foster sustainable practices have encouraged the pursuit of good health and well-being as one of the sustainable development goals (SDGs) [1,2]. In this context, resilient efforts to combat infectious diseases and pathogenic bacteria (such as *Bacillus cereus* and *Escherichia coli*) that cause foodborne illnesses and/or pose serious problems to human health is a current research quest [3]. For a time, the discovery of antibiotics was a milestone that proved effective towards the eradication of these bacterial pathogens. However, this breakthrough has been significantly watered down due to the rapid spread of antimicrobial resistance thus necessitating a need to search for better alternatives including natural antimicrobial agents from food sources [4,5].

Miang is an ethnic fermented tea made from leaves of *Camellia sinensis* var. *assamica* and has been reported to possess strong antioxidant, anti-inflammatory and antimicrobial activities [6–8]. Kodchasee et al. [9] and Unban et al. [10] have extensively studied the two processes adopted in the production of fermented Miang products including nonfilamentous fungi growth based (NFP)-Miang and filamentous fungi growth based (FFP)-Miang respectively, where they attributed the bioactivities of these teas to tannins, phenolic acids, catechin and its derivatives. Therefore, Miang extracts could be explored as an eco-friendly bid to prolong the shelf-life of food products as well as inhibit the growth of pathogenic bacteria.

Second only to the gastrointestinal gut, the oral cavity is home to diverse and abundant microbial communities (> 700 bacterial species) some of which are major players in disease pathogenesis especially, in events of oral Dysbiosis [11,12]. The translocation of oral microbes to the gastrointestinal tract (GIT) via enteral or hematogenous routes has been linked to colorectal cancer, irritable and inflammatory bowel diseases [11]. *Streptococcus mutans* is an aciduric and acidogenic dental pathogen implicated in dental caries and has closely evolved in association with the human host making it a model Gram-positive organism [13]. Together with other anaerobes and oral *Streptococci*, these pathogens organize into biofilms on dental surfaces and metabolize food remnants (fermentable sugars). The biofilm formation is mediated through sortase A enzyme which anchors surface proteins on bacterial cell walls to allow host attachment by *S. mutans* biofilms. Consequently, these events lead to the formation of dental plaque, demineralization of enamel, and leaching of teeth components [14,15]. Even though antimicrobials such as Chlorhexidine (CHX) are usually incorporated in mouth rinses as chemical anti-plaque agents that serve as adjuncts to the conventional mechanical plaque control strategies such as tooth-brushing and dental flossing [16,17], long term usage leads to bacterial resistance, teeth stains, erosion of the oral mucosa and taste alteration [18,19]. Moreover, tannins found in tea and numerous other plants have been reported to exert potent bacterial growth inhibition against Gram-positive and Gram-negative bacteria [20]. However, these compounds are known to be bitter and astringent, cause tooth stains/discolorations as well as bind histidine-rich proteins in the saliva to cause a moisture void that leads to dry mouth [21,22]. These drawbacks necessitate the need to explore other effective food products-derived antimicrobial alternatives that are preferably devoid of tannins, with an acceptable sensory, organoleptic and consumer-friendly properties that could find application as constituents of mouthwash.

In this study, the antimicrobial activities of acetone extracts including NFP Miang, FFP Miang, young (YTL) and mature (MTL) tea leaves (raw materials for NFP and FFP Miang

respectively) were evaluated against two Gram-positive (*Bacillus cereus* TISTR 747 and *Streptococcus mutans* DMST 18777) and Gram-negative (*Escherichia coli* ATCC 22595 and *Salmonella enterica* serovar Typhimurium TISTR 292) bacteria respectively. In addition, the tea samples were chemically treated to produce tannin-free extracts after which their antimicrobial activities were further investigated against *S. mutans* DMST 18777.

## Materials and methods

### Sample collection

FFP and NFP Miang samples were purchased from local producers in Pa-dang sub-district, Muang district, Phrae Province and Papae sub-district, Mae-Tang district, Chiang Mai Province Thailand, respectively. The starting materials for FFP and NFP Miang fermentation were mature tea leaves MTL (begins at the sixth leaf from the tea shoot) and fresh young tea leaves YTL (covers the second to the sixth leaf from the tea shoot) were also obtained from the above-mentioned locations.

### Preparation of Miang extracts

FFP Miang, NFP Miang, MTL and YTL were dried in a vacuum dryer (VD53 Binder Oven, Germany) for 24 h at 50 ˚C then ground into a powder with a blender respectively. 80% acetone (100 mL) was added to each sample (5g) and the extraction was carried out in a shaking incubator (Daihan Labtech Co., Ltd., Namyangju-City, Korea) set at 30 ˚C, 150 rpm for 1 h. The solution was filtered using a 125 mm Whatman filter paper and concentrated at 40 ˚C using rotary evaporator (N-1000 Eyela Rotary Evaporator; Tokyo, Japan). The crude extract was dissolved in 20 mL of deionized (DI) water (mg extract/mL), and then stored at -20 ℃ freezer until further use. For the antimicrobial tests, an aliquot of each extract was prepared and the pH was adjusted to neutral using 0.1 M NaOH.

### Bacterial strains and growth conditions

Bacterial strains were collected from Thailand Institute of Scientific and Technological Research (TISTR), Department of Medical Sciences Thailand (DMST) and American Type Culture Collection (ATCC) including *Bacillus cereus* TISTR 747, *Salmonella enterica* ser. Typhimurium TISTR 292, *Streptococcus mutans* DMST 18777 and *Escherichia coli* ATCC 22595. Three pathogens were cultivated in Mueller Hinton agar (HiMedia, India) at 37 ˚C while *S. mutans* DMST 18777 was cultivated in Trypticase soy medium (HiMedia, India) supplemented with 1% sucrose at 37 ˚C under microaerophilic condition.

### Determination of antibacterial activities of neutralized (pH 7) and non-neutralized (pH 4.0–4.6) Miang extracts

The standard agar well diffusion method as described by Hossain et al. [23] with slight modification, was used to measure the antibacterial activities of Miang extracts. Briefly, 4–5 isolated colonies were taken from each 18–24 h pure culture, for subsequent subculture overnight in 5 mL normal saline. Afterward, the culture was adjusted to 0.5 McFarland turbidity standard (~$1.0 \times 10^8$) and 200 µL of each bacterial culture was spread on agar plate surfaces. A sterile borer (6mm diameter) was used to make wells on the agar and 30 µL of the Miang extracts each or control (DI water) were dispensed into the well and the plates were incubated for 24 h at 37 ˚C. The zone of inhibition around each well was measured as diameters and the units were represented in millimeters (mm). Inhibition by neutralized extract and organic acids is calculated according to the following equation: inhibition by neutralized extract (%) = (zone of

inhibition for neutralized extract/zone of inhibition for non-neutralized extract) × 100, and inhibition by organic acids (%) = 100 - inhibition by neutralized extract.

## Determination of minimum inhibitory (MIC) and minimum bactericidal concentrations (MBC) of neutralized (pH 7) and non-neutralized (pH 4.0–4.6) Miang extracts

The broth dilution method was used to determine MIC and MBC according to the method of Nibir et al. [24] with little modifications. Briefly, overnight cultures were adjusted to 0.5 McFarland standards (~$1.0 \times 10^8$ CFU/mL) and a 1:1 dilution of each bacterial culture was inoculated into two-fold serially diluted Miang extracts respectively. After incubation for 18 h, 5 μL of the serially diluted cultures were plated onto Mueller Hinton agar and Trypticase soy agar (*Streptococcus mutans*) as drop plates and incubated for 24 h at 37 ˚C. The bacteriostatic (MIC) and bactericidal (MBC) effects for this study were considered as <3-log and ≥3-log decrease in CFU/mL, respectively, compared to the control inoculum [25,26].

## Total tannin contents of polyvinylpolypyrrolidone (PVPP)–Miang extract supernatants

The method of Makkar et al. [27] was adopted where the Miang extracts were chemically treated with Polyvinylpolypyrrolidone (PVPP) to separate tannins from other phenols. Briefly, 50 mg/mL PVPP was mixed with Miang extracts each, the mixtures were stirred for 15 mins and centrifuged afterward. The antibacterial activity of the PVPP-treated extract adjusted to pH 5 [28], was evaluated using the agar well diffusion method as described in 2.4 above. The total tannin (TT) content in the PVPP-treated extracts and original extracts respectively, were measured using Folin–Ciocalteu reagent [29] and the absorbance was recorded at 750 nm for all samples. All results were expressed as mg tannic acid equivalent per gram of sample (TAE/g).

## HPLC-QToF-MS analysis Polyvinylpolypyrrolidone (PVPP)–treated Miang extracts

The supernatants were analyzed using a Hitaci Chromaster HPLC system (Hitaci HighTech, Chiyoda-ku, Tokyo, Japan) coupled to a time-of flight mass spectrometer (Chromaster 5610 Q-TOF, Tokyo, Japan). The separation of the chemical compounds was conducted using a Pinnacle-II C18 column (250 × 4.6 mm, 5 μm) at a flow rate of 0.8 mL/min. The column temperature and detection wavelengths were set at 40 ˚C, 210 and 270 nm, respectively. 10.0 μL of each sample was injected. The mobile phase was composed of 0.1% acetic acid in water (v/v, A) and acetonitrile (B), with an elution of 95% A and 5% B. The mass spectrometer was operated in the positive ionization mode over a scan range of $m/z$ 50–1000 with the following settings: capillary voltage, 3.5 kV; counter gas flow, 0.6 L/min; nebulizer, 72.5189 psi; ionizing temperature, 2500 V.

## Inhibition of biofilm formation

*Streptococcus mutans* DMST 18777 was cultured overnight at 37 ˚C in *Trypticase soy* broth to attain an optical density at 600 nm (OD600) of ~1 [30]. The overnight culture was then diluted (1:10) in TSB *supplemented with* 1% sucrose and extracts' PVPP-treated extracts were added to desired concentrations respectively. 200 μL of the bacterial inoculum were aliquoted into wells of a 96-well microplate (flat-bottom) in the absence (untreated control) or presence of the supernatants and incubated at 37 ˚C for 24 h. After incubation, the medium was removed and

biofilms were washed with phosphate-buffered saline (PBS) twice. The plates were stained with 200 μL 0.1% aqueous crystal violet solution (CV) and incubated for 15 mins at room temperature. The plates were washed twice with water and the remaining CV was solubilized using 33% acetic acid (150 μL) and incubated under constant shaking for 5 mins. The absorbance was recorded at 595 nm and biofilm inhibition was estimated by calculating the absorbance at 595 nm of supernatant-treated wells as a percentage of the untreated control wells while the concentration for each supernatant at which 50% reduction in biofilm formation was observed when compared to the untreated wells was considered as the $IC_{50}$.

## Homology modelling

The target amino acid sequence of *S. mutans* sortase (WP_002269270.1) was retrieved form the National Center of Biotechnology Information (NCBI) database. The three dimensional (3D) structure of *S. mutans* sortase is available in the protein data bank (4tqX.pdb, www.rcsb.org). However, this template had 40 amino acid residues (predicted transmembrane domain) truncated from its N-terminal, residues 49–53 were omitted and active site amino acid (His139) was mutated to stabilize the template crystal structure [31]. The academic version of MODELLER10.4 was used to generate the 3D structure based on target-template alignment information (99% identity) using slightly modified python codes [32,33]. Multiple 3D models were generated but the best model with the lowest discrete optimized protein energy (DOPE) score was selected. The accuracy and stereochemical quality of the model (.pdb file) was evaluated by PROCHECK, ERRAT, Verify3D and WHATCHECK [34]. The model satisfied all the evaluation criteria thus was employed for molecular docking study.

## Molecular docking

The 3D structures of the identified compounds in the PVPP-treated extracts, were retrieved from PubChem database (https://pubchem.ncbi.nlm.nih.gov/) as 3D SDF conformer (accessed on July 4[th], 2023). To understand binding interactions between the compounds and sortase A (.pdb model), docking analyses were performed using Autodock Vina [35]. The.pdb file was prepared for docking by removal of all fluids and ions in the crystal structure, followed by the addition of hydrogen atoms, Gasteiger charges and protonation states to ionizable amino acids at physiologic conditions (pH 7.0). The 3D geometry of the ligands was converted from SDF format to PDBQT format and optimized via the Merck Molecular Force Field (MMFF) in open BaBel software. Using Autodock 4.2 [36]. The catalytic triad (His139, Arg213, and Cys205) were used to set up a grid box in order to cover the active site region whereas the center (12.158, 26.714, -14.504) and grid dimension (38, 52, 68) were generated in Autogrid. Molecular docking was carried out using the optimized ligands in Autodock Vina while transchalcone (co-crystalized with the template 4tqX), doxercalciferol, and biapenem were included in the docking analyses as reference sortase inhibitors for validation [31,37]. Results were clustered by positional root mean square deviation (RMSD) and represented by the lowest binding free energy (kcal/mol) followed by examination of binding interaction in Biovia discovery studio software [38].

## Statistical analysis

All experiments were carried out in triplicate and data are expressed as mean ± SD whereas one-way analysis of variance (ANOVA) was used to determine significant differences for multiple comparisons via SPSS software for Windows version 20 (SPSS Inc., Chicago, IL, USA). $p < 0.05$ was considered statistically significant.

**Table 1. The inhibition zone of Miang extracts tested against *Bacillus cereus* TISTR 747, *Escherichia coli* ATCC 22595, *Salmonella* Typhimurium TISTR 292 and *Streptococcus mutans* DMST 18777.**

| Sample | Zone of Inhibition | | | | | | | |
|---|---|---|---|---|---|---|---|---|
| | Non-neutralized extracts | | | | Neutralized extracts | | | |
| | YTL | MTL | NFP | FFP | YTL | MTL | NFP | FFP |
| pH | 4.3 | 4.6 | 4.3 | 4.0 | 7.0 | | | |
| *Bacillus cereus* TISTR 747 (mm) | 5.88 ± 0.63[c] | 3.00 ± 1.00[d] | 8.75 ± 0.25[a] | 7.00 ± 0.00[b] | - | - | 7.75 ± 0.75[b] | 5.25 ± 1.25[c] |
| *Escherichia coli* ATCC 22595 (mm) | - | - | 4.25 ± 0.75[b] | 5.50 ± 0.50[a] | - | - | 4.75 ± 0.25[b] | 2.75 ± 0.25[c] |
| *Salmonella* Typhimurium TISTR 292(mm) | - | - | 6.50 ± 0.50[c] | 6.00 ± 1.00[c] | - | - | 8.25 ± 0.75[a] | 7.75 ± 0.25[b] |
| *Streptococcus mutans* DMST 18777 (mm) | 4.00 ± 1.00[e] | 3.00 ± 0.50[e] | 13.75 ± 1.25[a] | 12.00 ± 0.50[b] | 4.50 ± 0.50[e] | 4.00 ± 0.00[e] | 9.00 ± 0.00[c] | 6.75 ± 0.25[d] |

YTL: Young tea leaves (109.5 mg/mL), MTL: Mature tea leaves (100.5 mg/mL), NFP: Non-filamentous fungi growth based process fermented Miang (74.5 mg/mL), FFP: Filamentous fungi-growth based process fermented Miang (91.5 mg/mL). Different alphabets represent significant differences ($p<0.05$, Tukey's test).

## Results

In this study, the antibacterial activities of original (pH 4.0–4.6) and neutralized (pH 7) Miang extracts were investigated against Gram-positive *Bacillus cereus* TISTR 747; *Streptococcus mutans* DMST 18777, and Gram-negative *Escherichia coli* ATCC 22595; *Salmonella enterica* ser. Typhimurium TISTR 292, respectively. Using the agar-well diffusion technique, NFP and FFP Miang extracts showed better antibacterial activities compared to young (YTL) and mature tea leaves (MTL). At neutral pH, NFP and FFP Miang still inhibited the growth of *B. cereus* TISTR 747 and *S. mutans* DMST 18777 with organic acids in NFP Miang contributing ~11.4% and 34.54% of the inhibition respectively (Table 1).

Similarly, ~25% and 43.75% of the antibacterial activity by FFP Miang against the Gram-positive pathogens was due to organic acids. On the contrary, neutralizing the extracts did not significantly ($p>0.05$) alter the antibacterial activity of NFP Miang against *E. coli* ATCC 22595 while FFP Miang showed ~50% antibacterial activity of the extract was as a result of tea polyphenols indicating bioactive acids constituents contributed the remaining 50% inhibition (Figs 1 and 2).

NFP and FFP Miang (pH 7.0) showed better growth inhibition against *Sm*. Typhimurium TISTR 292 compared to the original (non-neutralized) Miang extracts (Table 1). Antibacterial quantitative assays showed an overall better MIC (< 3 log decrease in CFU/mL) and MBC (> 3 log decrease in CFU/mL) for NFP and FFP Miang compared to the young (YTL) and mature tea leaves (MTL) against all the pathogens at non-neutralized and neutralized pH of the extracts respectively. Against *B. cereus* TISTR 747, an MBC of 109.5 mg/mL and 100.5 mg/mL was observed at non-neutral pH for YTL and MTL. On the contrary, the MBC observed at both pH of the extracts for NFP Miang (37.25 mg/mL) and FFP Miang (45.75 mg/mL) was significantly ($p<0.05$) lower when tested against the same pathogen (Fig 3A). YTL, MTL and FFP Miang showed significantly ($p<0.05$) better MICs (54.75 mg/mL, 25.13 mg/mL, 5.72 mg/mL) and MBCs (109.5 mg/mL, 50.25 mg/mL, 22.88 mg/mL) at non-neutralized pH against *E. coli* ATCC 22595 compared to the neutralized extracts. However, the MIC (9.31 mg/mL) and MBC (18.63 mg/mL) of NFP Miang when tested against the Gram-negative *E. coli* was the same irrespective of extracts' pH (Fig 3B). Moreover, YTL (non-neutralized) inhibited *Sm*. Typhimurium TISTR 292 with an observed MIC and MBC of 54.75 mg/mL and 109.5 mg/mL respectively, but no growth inhibition was observed for YTL and MTL at pH 7 nor for the normal (non-neutralized) extract of MTL (Fig 3C). On the other hand, NFP Miang had a similar MIC (9.31 mg/mL) and MBC (18.63 mg/mL) at both tested pH values whereas FFP Miang (at pH 7) showed a lower MIC (11.44 mg/mL) compared to the non-neutralized FFP Miang

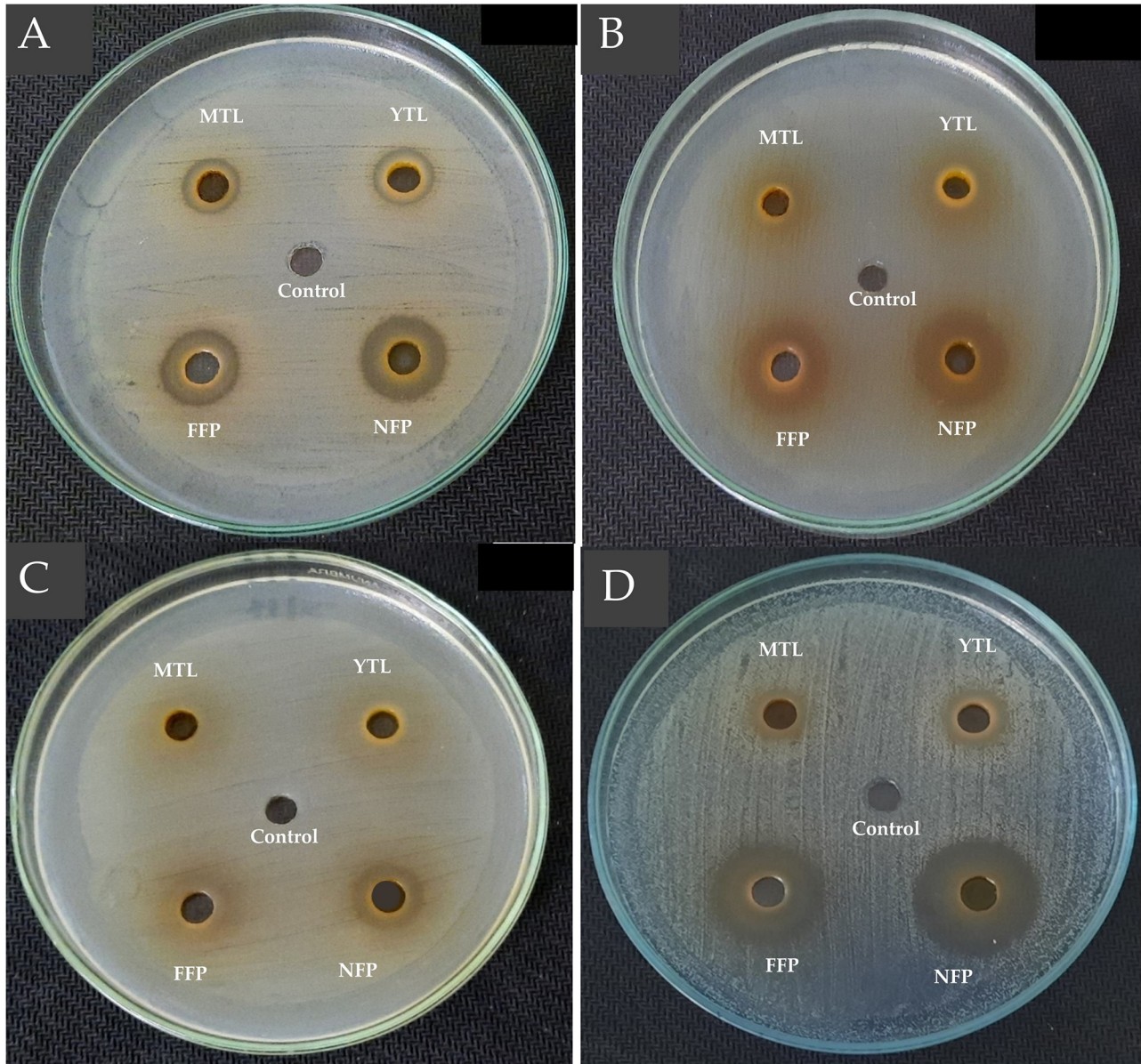

**Fig 1.** Antibacterial activities of non-neutralized tea leaf extracts and non-neutralized Miang extracts against *Bacillus cereus* TISTR 747 (A), *Escherichia coli* ATCC 22595 (B), *Salmonella* Typhimurium TISTR 292 (C), *Streptococcus mutans* DMST 18777 (D). Sterile water was used as the control. YTL: Young tea leaves, MTL: Mature tea leaves, NFP: Non-filamentous fungi growth based process fermented Miang, FFP: Filamentous fungi-growth based process fermented Miang (pH 4.0–4.9).

extract (22.88 mg/mL). The extracts had the best bacterial growth inhibition against *S. mutans* DMST 18777 with an MBC range of 3.42–6.84 mg/mL in YTL and MTL; while NFP and FFP Miang showed an MIC of 0.29 and 0.72 mg/mL respectively, albeit the MIC of NFP Miang (pH 7) was significantly ($p<0.05$) lower (0.15 mg/mL) compared to the non-neutralized extract (Fig 3D).

In this regard, tannin-free extracts were further prepared and tested against *S. mutans* DMST 1877 and the result showed a significant ($p<0.05$) difference in the zone of inhibition compared to the original (tannin-rich) extracts (Table 2).

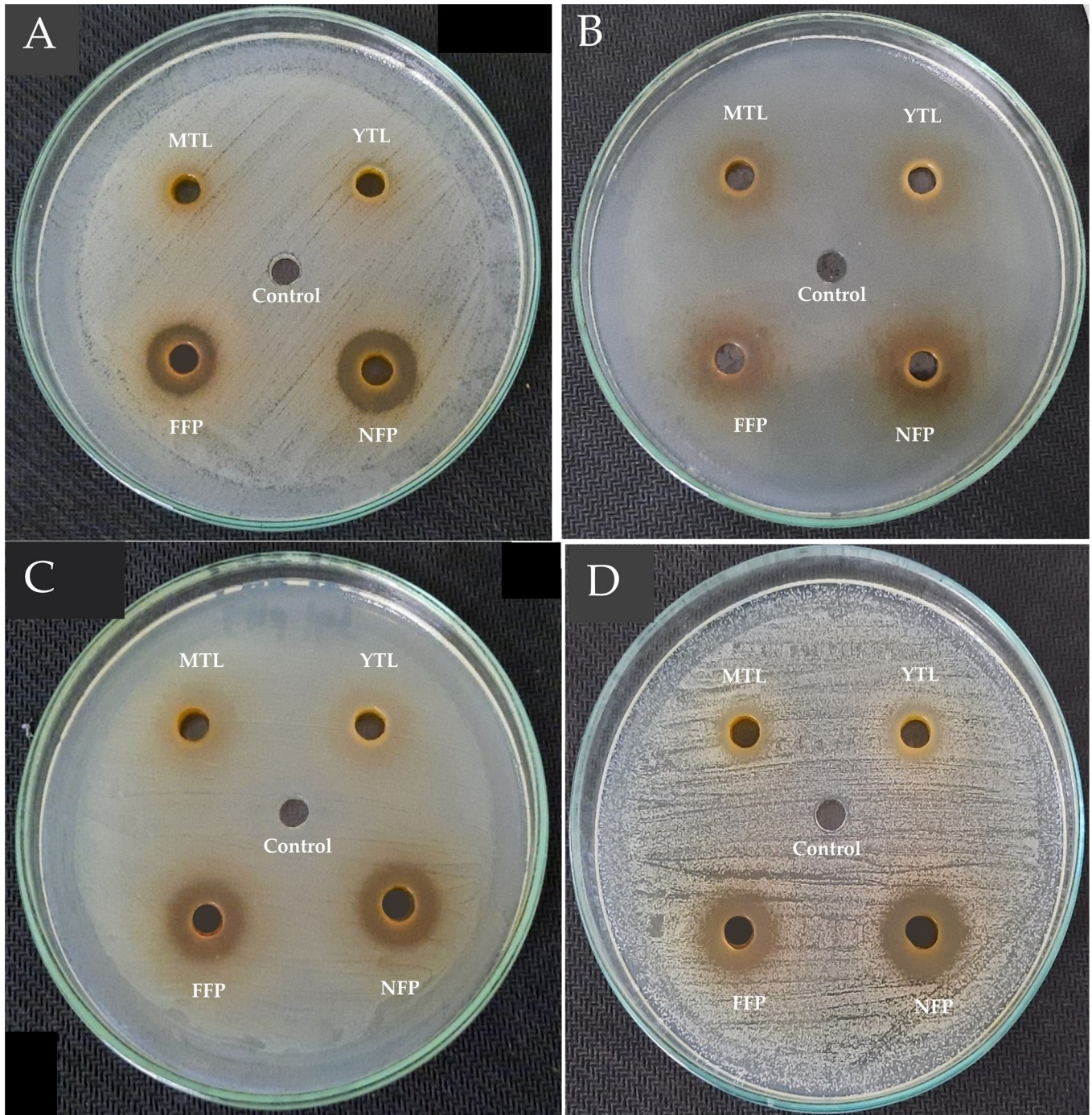

**Fig 2.** Antibacterial activities of neutralized tea leaf extracts and neutralized Miang extracts against *Bacillus cereus* TISTR 747 (A), *Escherichia coli* ATCC 22595 (B), *Salmonella* Typhimurium TISTR 292 (C), *Streptococcus mutans* DMST 18777 (D). Sterile water was used as the control. YTL: Young tea leaves, MTL: Mature tea leaves, NFP: Non-filamentous fungi growth based process fermented Miang, FFP: Filamentous fungi-growth based process fermented Miang (pH 7.0).

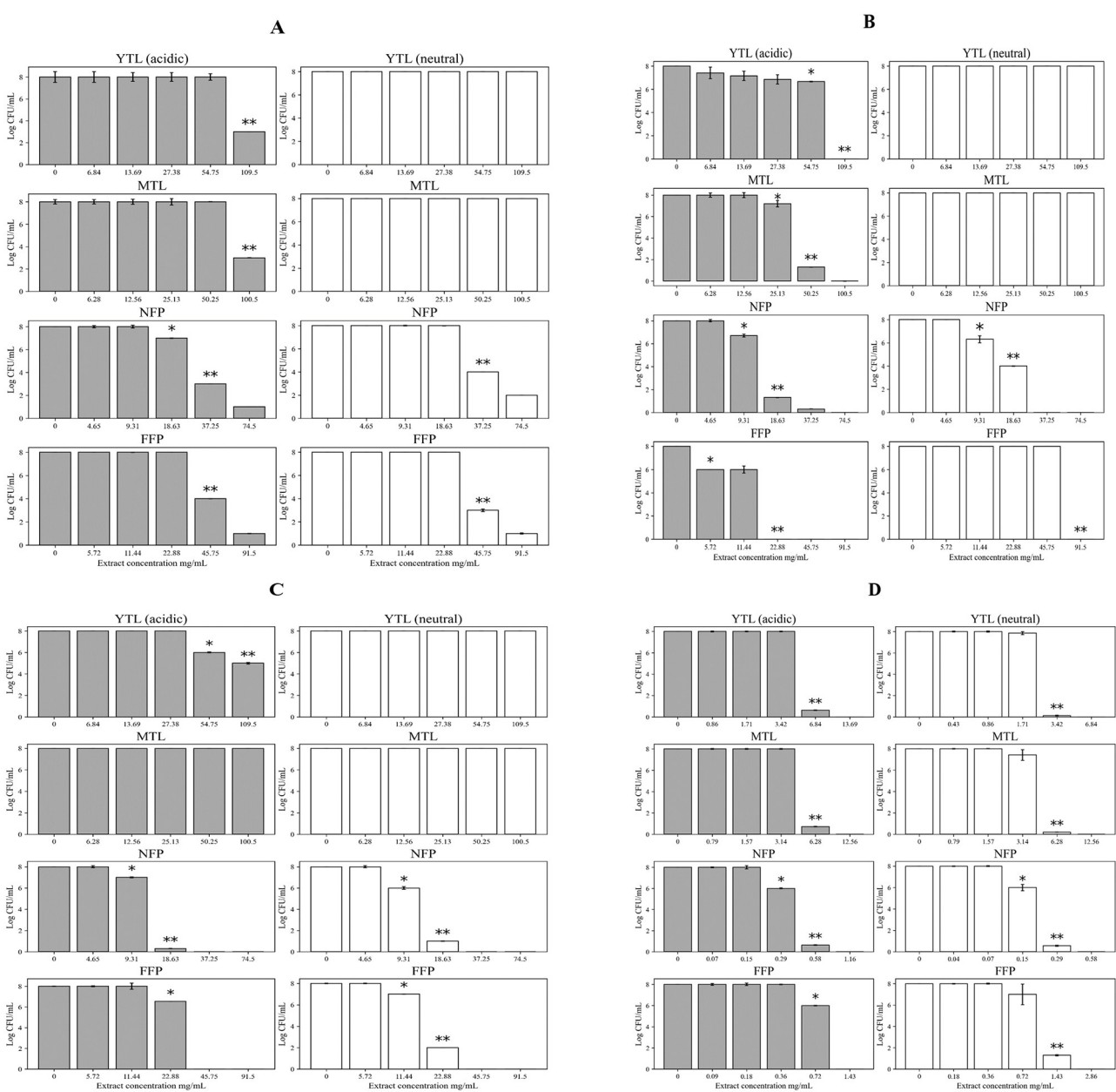

**Fig 3.** Antibacterial activities of non-neutralized and neutralized extracts of Miang and tea leaves against *Bacillus cereus* TISTR 747 (A), *Escherichia coli* ATCC 22595 (B), *Salmonella* Typhimurium TISTR 292 (C), *Streptococcus mutans* DMST 18777 (D). Young tea leaves (YTL), mature tea leaves (MTL), non-filamentous fungi-process based fermented Miang (NFP) and filamentous fungi-process based fermented Miang (FFP). "Different asterisk * and ** represent significant differences of MIC and MBC, respectively ($p<0.05$)".

However, tannin-free extracts still maintained potent inhibitions against *S. mutans* DMST 18777 (Fig 4). It was important to test the antibacterial activities of the extracts at non-neutralized and neutralized pH because it showed the observed activities was not exclusively due to low extract pH. Moreover, the pH of the tannin-free extract was adjusted to 5 which is still within the acceptable criteria for mouthwash (4.0–6.5) [28].

**Table 2. The inhibition zone of tannin-free extracts of Miang and tea leaves against *Streptococcus mutans* DMST 18777.**

| | SAMPLE | | | | | | | |
|---|---|---|---|---|---|---|---|---|
| | YTL | | MTL | | NFP | | FFP | |
| | Tannin-rich extract | Tannin-free extract | Tannin-rich extract | Tannin-free extract | Tannin-rich extract | Tannin-free extract | Tannin-rich extract | Tannin-free extract |
| *Streptococcus mutans* DMST 18777 (mm) | 15.00 ± 1.00[a] | 8.00 ± 0.00[b] | 13.00 ± 0.50[a] | 7.00 ± 0.50[b] | 18.50 ± 0.25[a] | 12.50 ± 1.00[b] | 22.50 ± 0.50[a] | 10.50 ± 0.25[b] |

YTL: Young tea leaves (109.5 mg/mL), MTL: Mature tea leaves (100.5 mg/mL), NFP: Non-filamentous fungi growth based process fermented Miang (74.5 mg/mL), FFP: Filamentous fungi-growth based process fermented Miang (91.5 mg/mL). Different alphabets represent significant differences ($p<0.05$, Independent samples T-test).

Considering the residual tannin contents after precipitation was significantly ($p<0.05$) low, this suggests that antimicrobial activities of the PVPP-treated extracts may be due to other compound classes (Fig 5).

Compound annotations obtained from LCMS analysis (S1 Table) shows the presence of gallic acid, ellagic acid, quinic acid, lactic acid and butyric acid, flavonoid aglycones and flavonoid glycosides in the fermented PVPP-treated extracts (S1 Fig). The inhibitory activity of PVPP-treated extracts against the formation of *S. mutans* DMST 18777 biofilms was also investigated. CV-staining showed better $IC_{50}$ values for NFP Miang (4.30 ± 0.66 mg/mL) and FFP Miang (12.73 ± 0.11 mg/mL) compared to the young and mature tea leaves respectively (Fig 6).

Molecular docking studies using the identified compounds against *S. mutans* sortase A showed stronger inhibition for ellagic acid, flavonoid aglycones and glycosides (Fig 7) which have been previously detected in NFP Miang and FFP Miang.

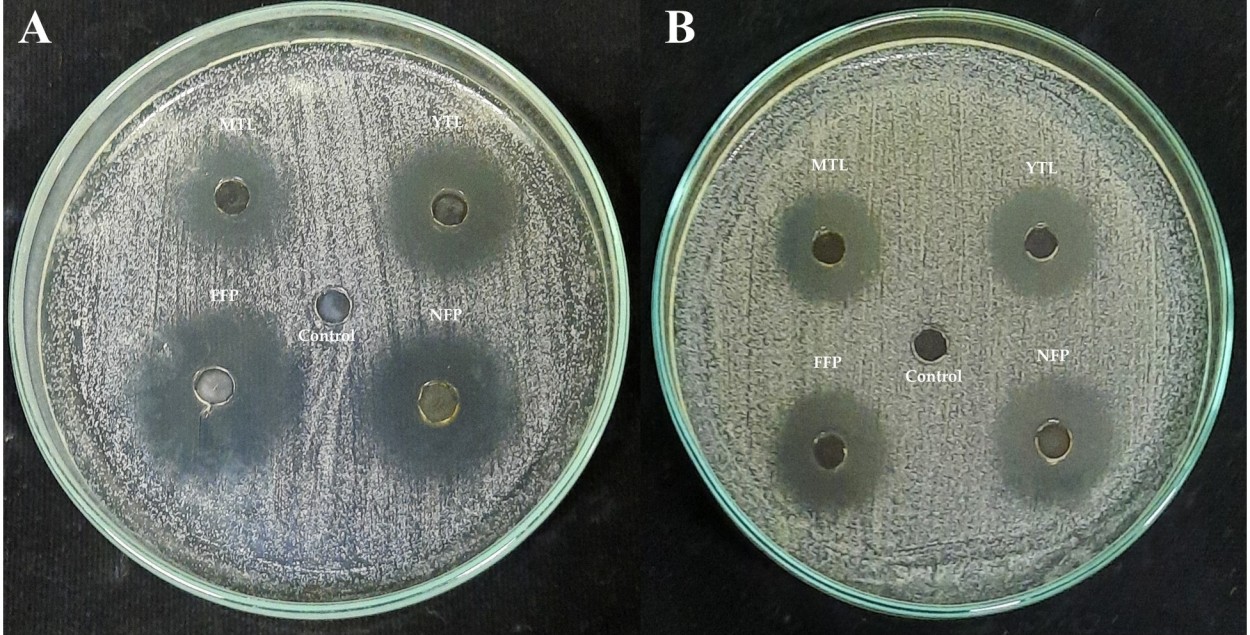

**Fig 4.** (A) Antibacterial activities of tannin rich and (B) tannin-free (pH 5) Miang extracts against *Streptococcus mutans* DMST 18777. YTL: Young tea leaves, MTL: Mature tea leaves, NFP: Non-filamentous fungi growth based process fermented Miang, FFP: Filamentous fungi-growth based process fermented Miang.

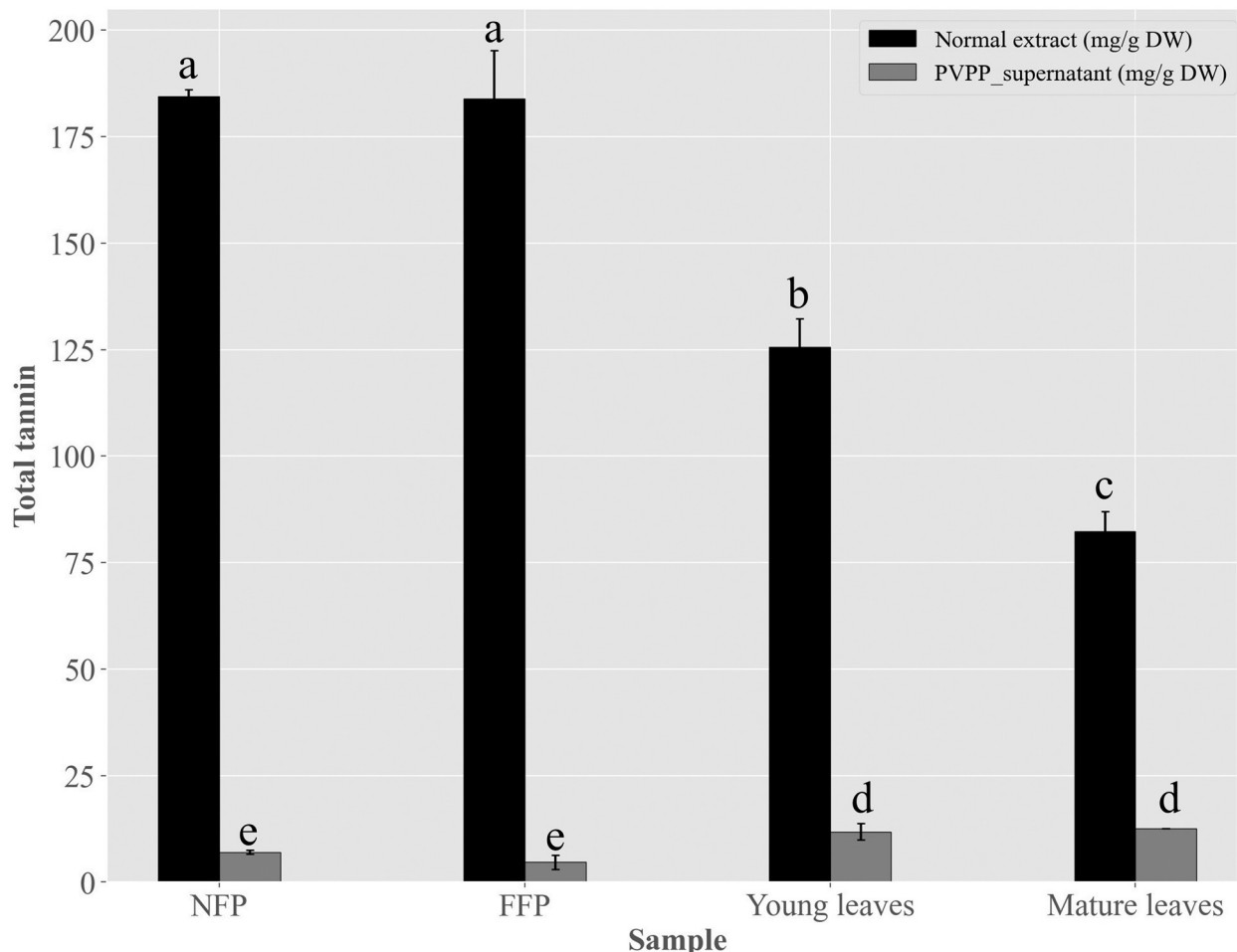

**Fig 5. Total tannin contents in Miang extracts before and after polyvinylpyrrolidone (PVPP)–precipitation.** Young tea leaves, mature tea leaves, non-filamentous fungi-process based fermented Miang (NFP) and filamentous fungi-process based fermented Miang (FFP). Different alphabets represent significant differences ($p<0.05$, Tukey's test).

Like the reference compounds (Fig 8), epicatechin, gallocatechin, chlorogenic acid, and 5-feruloylquinic acid interacted with the active site catalytic triad (Cys205, Arg213 and His139) suggesting a probable competitive-like inhibition (Table 3).

## Discussion

A cheaper, reliable, and sustainable bioprospection of plant extracts for possible application in innovative functional foods, natural active pharmaceutical ingredients (NAPIs) and/or therapeutics is a current research trend [39]. In this study, fermented Miang extracts (NFP and FFP Miang) showed better growth inhibition against both Gram positive (*B. cereus* TISTR 747 and *S. mutans* DMST 18777) and Gram-negative (*E. coli* ATCC 22595 and *Sm.* Typhimurium TISTR 292) pathogens compared to young (YTL) and mature (MTL) tea leaves respectively. The contents of tannins and organic acids including ellagic acid, galacturonic acid, tartaric acid, lactic acid, succinic acid, and acetic acid; in NFP and FFP Miang have been reported to significantly increase after fermentation [6] thus may have translated into the observed antimicrobial activities. The contribution of the compound classes to the antimicrobial activities was

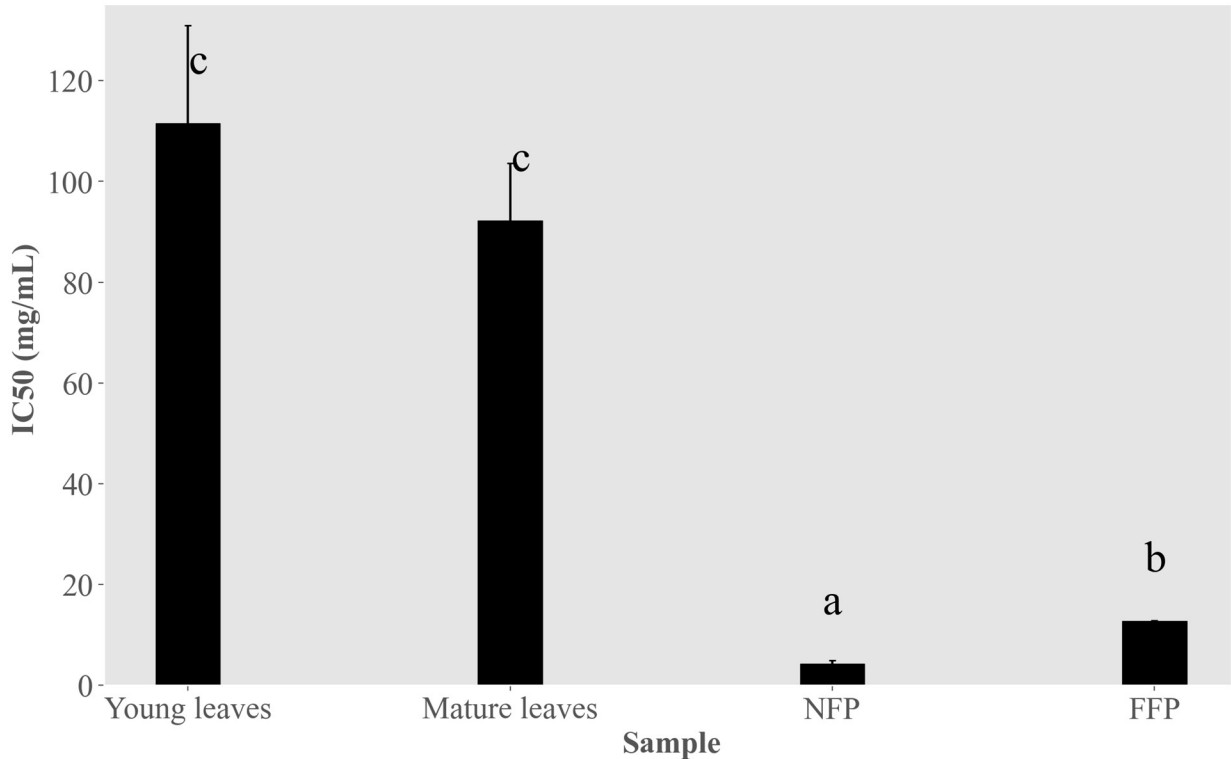

**Fig 6. Anti-biofilm activity of polyvinylpyrrolidone (PVPP)-treated Miang extracts on *S. mutans* DMST 18777.** Young tea leaves, mature leaves, non-filamentous fungi-process based fermented Miang (NFP) and filamentous fungi-process based fermented Miang (FFP). Different alphabets represent significant differences (*p*<0.05, Tukey's test).

investigated by neutralizing the pH of Miang extracts (Table 1) and the results suggested that organic acids and/or essential oils may actually confer significant antimicrobial activities in FFP Miang compared to NFP Miang especially, against *S. mutans* DMST 18777, *B. cereus* TISTR 747, and *E. coli* ATCC 22595. This observation for FFP Miang might be due the simultaneous stepwise fermentation process applied to produce the fermented product which uses filamentous fungi, yeast, and bacteria subsequently leading to microbial-mediated synthesis of acids, terpenes, esters, and other volatile aromatic compounds (VOCs) as these compounds have been previously characterized in FFP Miang using GC/MS analysis [9,40]. Some of these compounds including terpenes and alkaloids are known to permeate membranes and intercalate DNA in bacterial pathogens [41]. There are confirmed reports that show essential oils such as eugenol, in plant extracts contributed to potent growth inhibition of Gram-positive bacteria such as *S. mutans*. [42]. The zones of inhibition were larger for the fermented extracts tested under non-neutral pH (4.0–4.9) conditions against *S. mutans* DMST 18777 and *B. cereus* TISTR 747 (Figs 1 and 2) which suggests organic acids in NFP Miang and more so FFP Miang (S1 Fig), diffuse easier into the more permeable Gram-positive cell walls compared to the outer phospholipid membrane of Gram-negative bacteria which precludes cell invasion by lipophilic compounds [43,44]. Interestingly, FFP Miang (pH 7) showed a smaller zone of inhibition against *E. coli* ATCC 22595 while the same extract as well as NFP Miang (pH 7) extracts showed a better growth inhibition of *Sm*. Typhimurium TISTR 292 compared to their non-neutral counterparts (Table 1). The tannins and/or other non-acidic compounds alone are thought to be responsible for this inhibition against *Sm*. Typhimurium TISTR 292 at pH 7

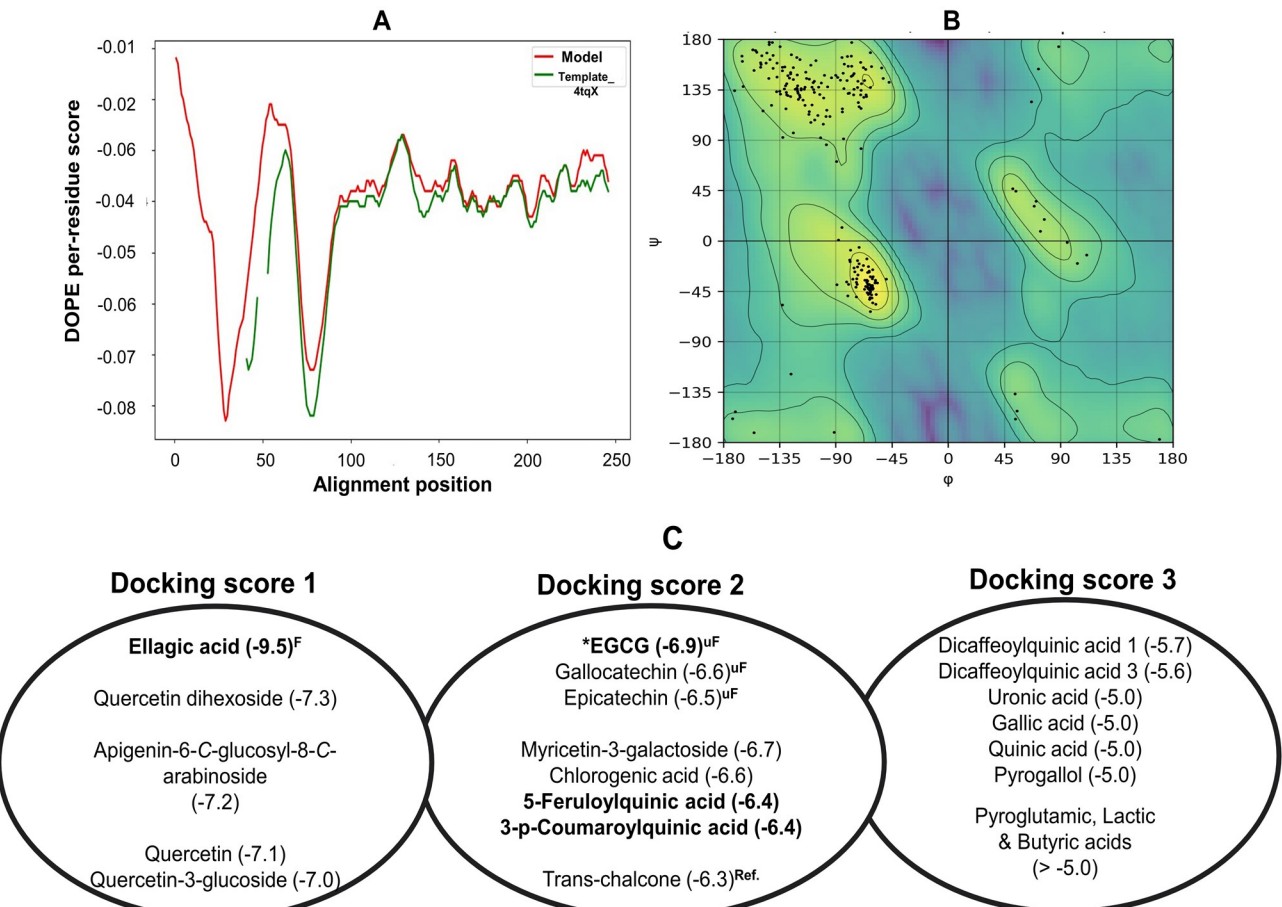

**Fig 7.** (A) Discrete optimized protein energy (DOPE) score profiles of the 4tqX template (-20008.758) and Model (-20812.996). The overlay shows the missing N-terminal residues in 4tqX template as well as the omitted residues (49–53). (B) Ramachandran plot generated via RamachanDraw for the best *Streptococcus mutans* sortase model (.pdb output file) from MODELLER10.4. (C) Binding free energies in kcal/mol for Compound-Sortase complex grouped in a descending order from left—right. **F**: Detected in fermented teas, **uF**: Detected in young and mature teas, **Ref.**: Reference compound.

which suggest the presence of potent compounds with salient pharmacophores capable of probable penetration and targeting of cellular machineries in this pathogen. In addition, catechins have been previously implicated in the disruption of bacterial membranes by interfering with the normal function of proteins on their cell surface [29,30]. However, the discrepancy observed for FFP Miang (at pH 7) against *E. coli* in this study could be further investigated in the future so as to shed more light into how the non-neutral and/or volatile compounds contained in FFP Miang exerted their growth inhibition against this Gram-negative pathogen.

Moreover, the inhibition pattern reported using agar well diffusion technique agreed with that observed for the quantitative experiments albeit the latter was able to give an idea as to the strength of extract inhibition through the MIC and MBC estimations. Overall, the fermented teas showed better MIC/MBCs compared to the young and mature tea extracts (NFP Miang > FFP Miang > YTL ~ MTL) against the bacterial pathogens with *S. mutans* DMST 18777 showing the best activity especially, for extracts tested at pH 7 (Fig 3D). On this note, the effect of PVPP-treated Miang extracts (tannin-free) were investigated against cariogenic *S. mutans* DMST 18777 since this pathogen is known to thrive on food/sugar remnants (such as sucrose & fructose) thereby causing infection by attaching to hosts' oral cavities using bacterial

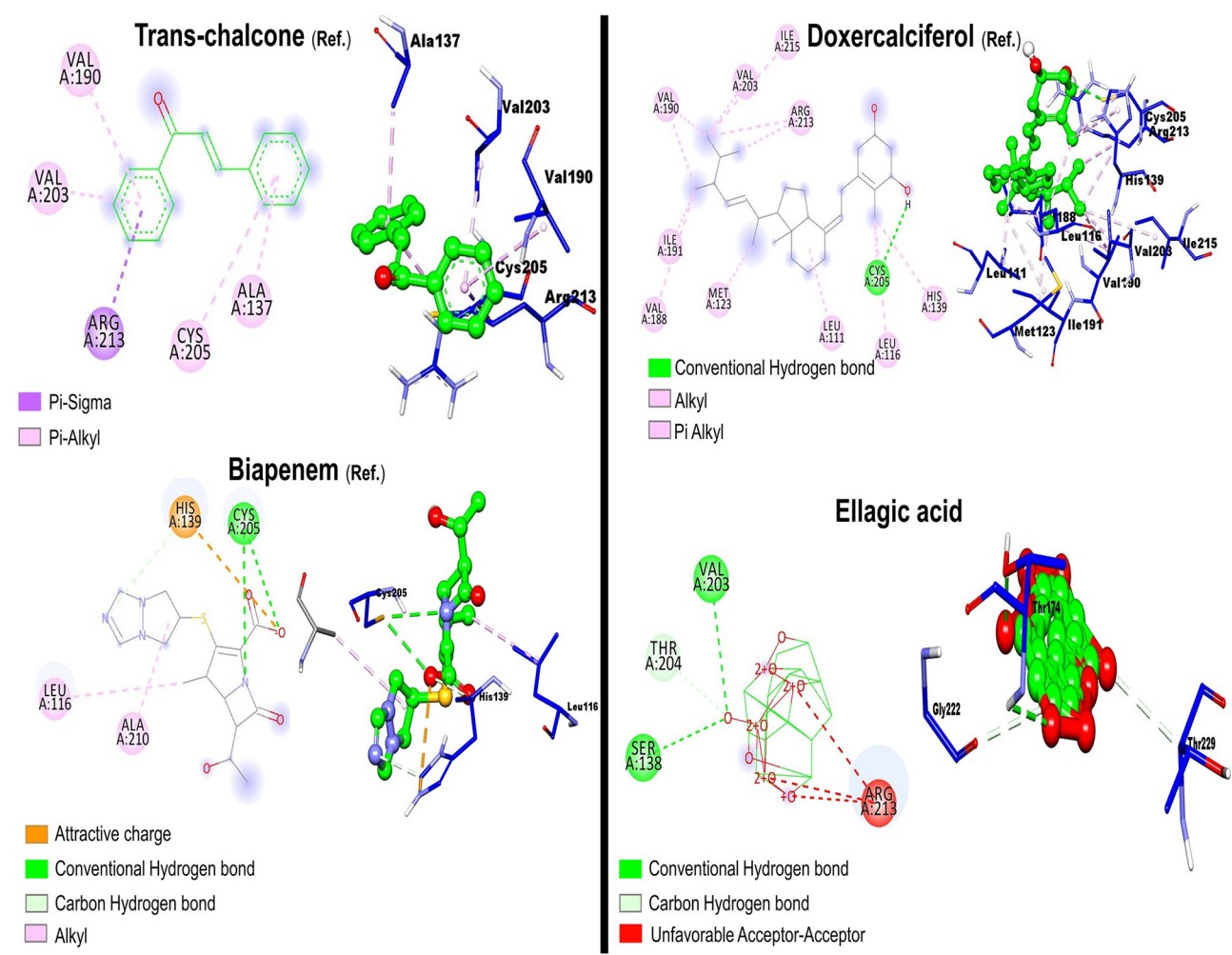

**Fig 8. 2D and 3D Ligand-receptor interactions for reference compounds and ellagic acid (bottom right).**

adhesins, exopolysaccharides amongst others, thus aggregating into effective and infectious biofilms that are less susceptible to antibiotics compared to planktonic pathogens [45–48]. *Streptococcus mutans* has been reported to recruit sucrose-dependent mechanisms where glucosyltransferase (GTF) mediates a reactions with glucan-binding proteins (GBP) to cause the formation of dental plaques [49–51]. Interestingly, tannins have been reported to inhibit GTF-mediated synthesis of insoluble glucans thus suppressed *S. mutans* proliferation [52]. Therefore, targeting these biofilms will require 2–4 folds the original MICs calculated for the test fermented extracts to be able to arrive at experimentally-relevant concentrations that could effectively inhibit the bacterial biofilms [53]. The lower MIC and MBC observed for the extracts at pH 7 (Fig 3D) also suggests a retained antimicrobial activity independent of organic acids which could make the extracts choice selections to be applied in therapeutic formulations targeted against aciduric and acidogenic *S. mutans* strains. However, 2–4 folds the original MICs might increase tannin contents to thresholds that might be toxic to the host system in addition to the sensory drawbacks of tannin deposition which causes tooth stains and bitter sensation thus will perhaps guarantee consumers' dissatisfaction with such therapeutic formulations [21].

**Table 3. Binding energies and bonding interactions of docked compounds.**

| # | Compound | Binding energy (kcal/mol) | Hydrogen bonding | Van der Waals[a], Hydrophobic & Electrostatic[b] interactions |
|---|---|---|---|---|
| Ref. | Trans-chalcone | -6.3 | - | Cys205, Ala137, Val203, Val190, Arg213. |
| Ref. | Biapenem | -6.0 | Cys205, His139 | His139[b], Ala210, Leu116 |
| Ref. | Doxercalciferol | -6.8 | Cys205 | His139, Leu116, Leu111, Arg213, Val203, Ile215, Val190, Ile191, Val188, Met123. |
| 1 | Ellagic acid | -9.5 | Thr204, Ser138, Val203. | Arg213* |
| 2 | Quercetin dihexoside | -7.3 | Gln239, Ser240, Ser245, Gln243, Phe246. | Lys102. |
| 3 | Apigenin-6-*C*-glucosyl-8-*C*-arabinoside | -7.2 | Ser240, Phe246, Gln243, Lys102, Asn242. | Tyr241[a], Lys102, Pro99. |
| 4 | Quercetin | -7.1 | Ile103, Tyr241. | Lys102. |
| 5 | Quercetin-3-glucoside | -7.0 | Ser245, Tyr241, Gly146. | Lys102, Gln149. |
| 6 | Epigallocatechin gallate | -6.9 | Tyr241, Gln243, Phe246, Gln239. | Lys102. |
| 7 | Myricetin-3-galactoside | -6.7 | Asn193, Asp192, Val190, Asn133, Gu199, Glu189, Lys181. | His187, His217, Arg194. |
| 8 | Chlorogenic acid | -6.6 | Cys205, Ser138, His187, Pro185. | Arg213[a], Val203[a], His139[a], Ala137, Leu116. |
| 9 | Gallocatechin | -6.6 | Cys205, Arg213, His187, Val203, Ser138. | Ala137, Met123. |
| 10 | Epicatechin | -6.5 | His187, Arg213, Cys205. | Leu111[a], Ile119[a], Val188[a], Met123, Val203, Ala137. |
| 11 | 5-Feruloylquinic acid | -6.4 | Cys205, Pro185, Thr184. | Ala137, Met123[a], Val203[a]. |
| 12 | 3-p-Coumaroylquinic acid | -6.4 | Tyr241, Gln243, Ser245. | Lys102. |
| 13 | Dicaffeoylquinic acid 1 | -5.7 | Gln239. | Phe246. |
| 14 | Dicaffeoylquinic acid 3 | -5.6 | Glu156, Ser148, Gly146, Gly143, Met144, Ser147. | Thr206[a], Glu156, Phe142. |
| 15 | Gallic acid | -5.6 | Arg194, Val190, Glu189. | His217. |
| 16 | Uronic acid | -5.6 | Phe246, Ser245, Gln243. | Lys102 |
| 17 | Pyrogallol | -5.0 | Asp192, Val190. | Arg194, His217. |
| 18 | Quinic acid | -5.0 | Lys181, Glu189, Glu179. | - |
| 19 | Pyroglutamic acid | -4.6 | Thr184, Pro185, Val188. | - |
| 20 | Lactic acid | -4.1 | Asn132, Asn193, Lys124. | - |
| 21 | Butyric acid | -3.7 | Ile103. | - |

*represents an unfavorable positive-positive interaction.

To address this, PVPP-treated extracts tested against *S. mutans* DMST 18777 using agar well diffusion technique showed residual antimicrobial activity after precipitating out tannins (Table 2) more so for NFP Miang compared to the other extracts (NFP Miang > FFP Miang > YTL ~ MTL) (Fig 4). The efficiency of tannin precipitation was evaluated by quantitative estimation of tannin contents in the original Miang extracts and their corresponding PVPP-treated extracts where the result showed a > 90% precipitation efficiency overall (Fig 5) which mildly suggest other compound classes and/or the residual tannins (in young and mature teas) might be responsible for the growth inhibition of this supernatant against *S. mutans* DMST 18777. Interestingly NFP Miang (4.30 mg/mL; $IC_{50}$) and FFP Miang (12.73 mg/mL; $IC_{50}$) supernatants showed appreciable inhibitory activities against *S. mutans* DMST 18777 biofilm formation which indicates that the antibacterial activity of the tannin-free supernatants from these fermented samples were potent enough to exert anti-biofilm activities. A computational docking study was considered given the effectiveness of the PVPP-treated extracts in the inhibition of the Gram-positive *S. mutans* DMST 18777. Sortase A is a highly conserved membrane-associated enzyme in Gram-positive bacteria that facilitate the covalent

attachment of bacterial virulent-surface proteins on bacterial cell walls to mediate host cell attachment, biofilm formation, iron acquisition, signaling, invasion and pili formation [31,54,55]. Sortase A recognizes the conserved LPXTG (Leucine, proline, X = any amino acid, threonine, glycine) motif at the C-terminus of *S. mutans* surface proteins and hydrolyzes the amide bond between threonine and glycine residues using a conserved active site cysteine (Cys205) residue. The thioacyl intermediate formed attaches to the cell wall precursor lipid II pentaglycine side chain. On the other hand, side chains of His139 and Arg213 in the active site serve as a general base as well as stabilizer of ionic intermediates in the sortase catalyzed hydrolysis thus making sortase A an attractive chemotherapeutic target [31,56–58]. Medicinal plants rich in flavonoids (including kempferol, apigenin and quercetin) and tannins (such as ellagitannins and proanthocyanidins) have been shown to contribute to the maintenance of oral health and are therefore applied as additives or major ingredients in toothpastes or mouthwashes [59]. Our results suggested ellagic acid, flavonoid aglycones, flavonoid glycosides and organic acid derivatives might be responsible for the antimicrobial activities of NFP and FFP Miang (Table 3, Fig 7C) from the observed docking binding energies. Further experiments could explore possibilities of enriching and/or optimizing tannin-free Miang extracts as potential candidates to be applied as natural active pharmaceutical ingredients (NAPIs).

## Conclusion

This study confirmed that microbial fermentation increased the antimicrobial activities of neutralized and non-neutralized Miang extracts against both Gram-positive (*B. cereus* TISTR 747 and *S. mutans* DMST 18777) and Gram-negative (*E. coli* ATCC 22595 and *Sm*. Typhimurium TISTR 292) bacteria. The quantitative estimation of antibacterial activities showed a lower MIC and MBC for all especially, the fermented extracts tested against *S. mutans* DMST 18777. The extracts were precipitated using PVPP and the tannin-free extracts still retained appreciable growth inhibition against *S. mutans* DMST 18777 while NFP and FFP Miang showed appreciable IC50 values against *S. mutans* DMST 18777 biofilms. Overall, the result showed the potential of these extracts to be enriched, optimized and possibly find future applications as a mainstay in combating bacterial pathogens and as potential ingredients in mouthwashes.

## Supporting information

**S1 Fig. HPLC chromatogram of polyvinylpyrrolidone (PVPP)–treated Miang extracts.** YTL: Young tea leaves, MTL: Mature tea leaves, NFP: Non-filamentous fungi growth based process fermented Miang, FFP: Filamentous fungi-growth based process fermented Miang. (PDF)

**S1 Table. Identification of compounds in polyvinylpyrrolidone (PVPP)–treated Miang extracts.** [a]Abbreviations are listed as follows: OA; organic acid; F3, flavan-3-ol; FL, flavonol/flavone; PA, phenolic acid; PP, Polyphenol; AA, amino acid. [b]RT: Retention time All acquisitions were carried out in the positive mode ($m/z$, [M + H]$^+$) using LC-MS (at 210 and 270 nm wavelengths). (PDF)

## Acknowledgments

The authors would like to acknowledge the Chiang Mai University's Graduate school and Presidential scholarship program, respectively. We also acknowledge Faculty of Agro-Industry, Chiang Mai University, for providing the research facilities.

## Author Contributions

**Conceptualization:** Chalermpong Saenjum, Chartchai Khanongnuch.

**Data curation:** Aliyu Dantani Abdullahi, Pratthana Kodchasee.

**Formal analysis:** Aliyu Dantani Abdullahi.

**Funding acquisition:** Chartchai Khanongnuch.

**Investigation:** Pratthana Kodchasee, Napapan Kangwan, Hathairat Thananchai, Chartchai Khanongnuch.

**Project administration:** Chartchai Khanongnuch.

**Resources:** Kridsada Unban.

**Supervision:** Kridsada Unban, Chalermpong Saenjum, Napapan Kangwan, Hathairat Thananchai, Kalidas Shetty.

**Writing – original draft:** Aliyu Dantani Abdullahi.

**Writing – review & editing:** Aliyu Dantani Abdullahi, Kalidas Shetty, Chartchai Khanongnuch.

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
