## [Decision Letter · Decision Letter 0]

7 Feb 2024

PONE-D-23-35234

Antibacterial activities of Miang extracts against selected pathogens and the potential of the tannin-free extracts in the growth inhibition of Streptococcus mutans

PLOS ONE

Dear Dr. Khanongnuch,

Thank you for submitting your manuscript to PLOS ONE. After careful consideration, we feel that it has merit but does not fully meet PLOS ONE’s publication criteria as it currently stands. Therefore, we invite you to submit a revised version of the manuscript that addresses the points raised during the review process.

We look forward to receiving your revised manuscript.

Kind regards,

Mozaniel Santana de Oliveira, Ph.D

Academic Editor

PLOS ONE

Journal Requirements:

Additional Editor Comments:

Dear Dr. Khanongnuch, I have made a decision about your manuscript, please review each reviewer's comment point by point, changes should be highlighted so we can track. 

Reviewers' comments:

Reviewer's Responses to Questions

**Comments to the Author**

1. Is the manuscript technically sound, and do the data support the conclusions?

Reviewer #1: Yes

Reviewer #2: Yes

Reviewer #3: Partly

Reviewer #4: Partly

Reviewer #5: Yes

2. Has the statistical analysis been performed appropriately and rigorously? 

Reviewer #1: Yes

Reviewer #2: I Don't Know

Reviewer #3: Yes

Reviewer #4: Yes

Reviewer #5: Yes

3. Have the authors made all data underlying the findings in their manuscript fully available?

Reviewer #1: Yes

Reviewer #2: Yes

Reviewer #3: Yes

Reviewer #4: Yes

Reviewer #5: Yes

4. Is the manuscript presented in an intelligible fashion and written in standard English?

Reviewer #1: Yes

Reviewer #2: Yes

Reviewer #3: Yes

Reviewer #4: No

Reviewer #5: Yes

5. Review Comments to the Author

Reviewer #1: The manuscript PONE-D-23-35234 entitled “Antibacterial activities of Miang extracts against selected pathogens and the potential of the tannin-free extracts in the growth inhibition of Streptococcus mutans” has described solvent extraction of Miang (fermented tea leaves) and tea leaves, raw material for Miang production and determination of antibacterial activities of the obtained extracts, and those treated with tannin-free against selected bacterial pathogens, examination of compounds. The results have found that tannin-free Miang extracts showed more potential to inhibit growth of pathogens, specifically Streptococcus mutans than tannin-free tea leaf extracts. In addition, the manuscript has described analysis of the extracts by HPLC-MS in order to examine the chemical compounds that are relevant to growth inhibition of Streptococcus mutans. The identified compounds were used to determine interactions between each identified compound and sortase A enzyme, the key enzyme associated with biofilm formation of S. mutans. The authors conclude that the tannin-free Miang extracts could be used as a natural active pharmaceutical ingredient for applications in oral hygiene products.

After the evaluation, this manuscript presents sufficient experimental designs and statistical analysis. However, there is unclear information and errors. To meet the standard manuscript for publication, the current form of the manuscript must be substantially revised. The following comments and suggestions would help increase the completement of the manuscript.

General comments and suggestions

Please check the writing format of the scientific name of all microorganisms. There are “Streptococcus” and “Salmonella” which can be abbreviated as “S.”. This would make readers confused. It is suggested to present “S.” for Streptococcus mutans and “Sm”. for Salmonella.

To describe Salmonella enterica serovar Typhimurium, it is recommended to state it as “Salmonella var. Typhimurium” or “Salmonella Typhimurium” or “Sm. Typhimurium”. Please correct it in the whole manuscript.

Please indicate strain to all microorganisms used in this study.

Full name of a microorganism must be present for the first time of the main section and then it should be abbreviated. However, when the authors state it as a subject of the sentence, it is recommended to state its full name.

It is confusing when the authors state NFP and FFP as Miang because either NFP or FFP is a process of Miang fermentation. It is recommended to state it as “NFP Miang” or “FFP Miang”. Please also correct these words throughout the manuscript.

Comma symbol (,) must be added after a specific word such as respectively, especially, etc.,

Considering the type of extracts used in this study and their consistency, please reconsider these words: non-neutral, neutral, acidic, and pH 7. If tea leaf extracts (YTL and MTL) and NFP and FFP Miang extracts are generally acidic, it could state them as “non-neutralized extracts”. After the authors have neutralized the acidic extracts, it could state them as “neutralized extracts”.

Due to Miang is made from tea leaves, it is relevant to express only young tea leaves or mature tea leaves rather than unfermented young tea leaves and unfermented mature tea leaves, respectively (Line 228 and others in discussion section).

Please italicize p of p-value and t of t-value in whole manuscript.

It is more relevant to replace “PVPP-treated extract’ supernatant” by “PVPP-treated extract”. (lines 148, 398, 408, …)

Major and minor comments

Introduction section, add more information about role of sortase A enzyme in dental plaque formation. This information would at least let readers primarily understand why the authors chose molecular docking to investigate interactions between active compounds in Miang extracts and the sortase A enzyme.

Line 109-107 how did the authors express the unit of all extract? Is it mg/mL total polyphenol content or mg/mL total tannin content, or mg extract/mL, etc., Please add more information.

Line 127, is it non-neutralized Miang extracts?

Line 139, is it neutralized Miang extracts? Please also consider as above mentioned.

Line 133, indicate diameter of borer.

Line 134, indicate temperature.

Line 163, remove “be”

Line 172, edit to “Streptococcus mutans” when it was first used as the subject.

Line 173, edit to “an optical density at 600 nm (OD600)”

Liness 223, 247, and 253 add the strain of S. mutans.

Lines 230 and 239 how did the authors calculate percentages of inhibition? Please clarify.

Table 1, the authors can categorize the extracts into two groups including non-neutralized (acidic) and neutralized extracts. Each group consists of YTL, MTL, NFP, and FFP.

Line 236-237, different alphabets represent significant differences vs control. The control was just water. If the authors compared the experimental values to the control, extract samples exhibited the clear zone will have significant differences. From the point of view, multiple comparisons were made after ANOVA results showed significant differences (p < 0.05). It is supposed that all pairwise comparisons were used to analyze the data within the same row, not just compared with the control. It is noted that data with positive results are always significant when they are compared with the control. Please consider this comment for Fig 3., Table 2, Fig 5, and Fig 6.

Lines 245 and 251, it is recommended to revise the figure legend as follows:

Fig 1. Antibacterial activities of non-neutralized tea leaf extracts and non-neutralized Miang extracts against Bacillus cereus TISTR 747 (A), Escherichia coli ATCC 22595 (B), Salmonella Typhimurium TISTR 292 (C), Streptococcus mutans DMST 18777 (D). Sterile water was used as the control.

Fig 2. Antibacterial activities of neutralized tea leaf extracts and neutralized Miang extracts against Bacillus cereus TISTR 747 (A), Escherichia coli ATCC 22595 (B), Salmonella Typhimurium TISTR 292 (C), Streptococcus mutans DMST 18777 (D). Sterile water was used as the control.

Lines 256-277, it is recommended to initially describe line 257-260 (Antibacterial quantitative assays showed …….at acidic and neutral extracts’ pH respectively). However, the phrase “at acidic and neutral extracts’ pH respectively” should be revised to proper format as previously suggested. For example, Antibacterial quantitative assays showed an overall better MIC and MBC for NFP and FFP Miang extracts compared to the YTL and MTL when they were tested against all the pathogens, respectively.

Line 278, it is recommended to revise the figure legend as follows:

Fig 3. Antibacterial activities of non-neutralized and neutralized extracts of Miang and tea leaves against Bacillus cereus TISTR 747 (A), Escherichia coli ATCC 22595 (B), Salmonella Typhimurium TISTR 292 (C), Streptococcus mutans DMST 18777 (D).

Line 282, Different asterisk (*) represent significant differences vs control. What was the control of this experiment? Is it the initial cell concentration or just sterile water? This phrase could be modified to “Different asterisk symbols * and ** represent significant differences of MIC and MBC, respectively (p < 0.05)”.

Line 289, please consider modifying description of table 2 as follows:

Table 2. The inhibition zone of tannin-free extracts of Miang and tea leaves against Streptococcus mutans DMST 18777. Avoid extension of figure description.

Line 293 the comment is similar to that is given for Line 236-237

Line 296, it is unnecessary to add Fig 4 because the authors have already displayed the results in Table 2.

Lines 309-314 it is necessary to compare effect of tannin-free extracts and their original extract (in the presence of tannin)

Lines 384-386 Streptococcus mutans has been reported to recruit sucrose-dependent mechanisms where glucosyltransferase (GTF) reacts with sucrose and convert into dextran polymer or glucan that can be further bound with glucan-binding proteins (GBP) which leads to the formation of dental plaque. Please consider modifying the sentence.

Lines 411-425 please shorten principal discussion about role of sortase A in plaque formation but discuss more about finding of this study. How could active compounds from PVPP-treated Miang extracts inhibit sortase A? Are there any literatures reported active compounds that have potential to inhibit the enzyme?

Line 438-440 the sentence in the conclusion is not relevant to what the author proposes in introduction section (Lines 51-53, and 90-93).

Reviewer #2: It was advantageous to discover a plant extract that inhibited pathogen growth. Several concerns, however, needed to be elucidated in order to have a better knowledge of Miang extracts' antibacterial activity.

It was unclear how the authors obtained the quantities of Miang extracts utilized in Table 1 to establish zones of inhibition against bacteria.

Please double-check whether it was Turkey's or Tukey's test (line 237).

Only triplicate samples used to run a t-test comparing tannin rich extract to tannin-free extract were sufficient to accept the results (Table 2). Please specify the concentration of Miang extracts that were used.

Please modify the titles of all figures to improve comprehension.

In the conclusion section, the authors should summarize the optimal concentration of each Miang extract that can be employed to inhibit bacterial growth based on the MIC:MBC ratio of Miang extracts.

Reviewer #3: Materials and Methods

1. Line 126 to 136 talks about ‘Determination of antibacterial activities of neutralized (pH 7) and non-neutral (pH 4.0 – 4.6) Miang extracts’. The description, however, misses out on how these extracts were obtained before using them for the microbial tests. The ‘Preparation of Miang extracts’ is also silent on how the neutral and non-neutral extracts were obtained. Authors should kindly take a look at that.

2. What was the rationale for testing neutralized and non-neutralized extracts in the study? Could the justification be clearly demonstrated in the write-up?

3. Line 127 – I guess ‘non-neural’ should be ‘non-neutral’.

4. Kindly take a second look at the format for writing your units. At one point, I see CFU/mL, then at other places, I see CFU mL-1.

5. Line 163-164 : authors should kindly the sentence ‘The separation of the chemical compounds was be conducted……..’

Results

1. From Table 2, the zone of inhibition for tannin-rich and tannin-free extracts against Streptococcus mutans were significantly different for each type of sample. How do you authors then arrive at the observation that ‘tannin-free NFP and FFP extracts had significantly (p> 0.05) comparable zones of inhibition with the tannin-rich extracts’ and refer to figure 4, which I suppose is the primary source of data for Table 2?

2. Based on the observations in Table 2, it would be appreciated if authors could reconsider the point on tannin-free producing similar effects as tannin-rich extracts. That obviously also affect the reasoning that

Reviewer #4: This article explain on the antibacterial activity of Miang extract againts selected foodborne pathogens and the finding showed that the extract had the ability to inhibit the growth of pathogens. However, there is no problem statement regarding foodborne pathogens eventhough at the the author mention about foodborne problem at the beginning. Author is suggested to do proof read on their article content. This research give new info on the antibacteria activity of miang extract

Reviewer #5: I extend my appreciation to the authors for their dedication and commendable efforts in completing this excellent piece of research. This work is of considerable importance within the context of oral health, addressing a notable gap in the current literature and searching for novel antimicrobial agents is of great interest. I recommend its publication in PLOS for the following reasons:

• The title effectively reflects the study, and the abstract is clear and concise for readers.

• The comprehensive literature review demonstrates a good understanding of existing research, particularly in the quest for alternative antimicrobial agents.

• The methodology is well-structured, offering a clear and systematic approach to the research, which can serve as a valuable guide for future researchers. I just suggest standardizing the writing style concerning bacteria.

• The data analysis is well-performed, and the presentation is clear.

• The discussion is thoughtful, providing significant interpretations of the results.

• The author displays a high level of proficiency in English.

• The use of references is appropriate.

Overall, this work makes a noteworthy contribution to the field in the search for a new antimicrobial agent and aligns well with the standards of PLOS. I strongly recommend its publication.

6. PLOS authors have the option to publish the peer review history of their article (what does this mean?). If published, this will include your full peer review and any attached files.

Reviewer #1: No

Reviewer #2: No

Reviewer #3: No

Reviewer #4: No

Reviewer #5: No

---

## [Author Response · Author response to Decision Letter 0]

4 Mar 2024

POINT BY POINT RESPONSE TO THE EDITOR AND REVIEWERS’ COMMENTS ON THE MANUSCRIPT (PONE-D-23-35234).

Thank you for giving us the opportunity to revise the manuscript and the willingness to accept the article after the major revision. We have considered each of the comments given by both editor and reviewers and have provided revisions/responses to each comment in a point-by-point pattern. Please note that all corrections by the authors are made using Tracked changes and saved as “Revised document with Track changes”. Line numbers (LNs) are also included in the point by point responses to reviewers for efficient tracking of corrections.

Editor

General Response: We are grateful to the editor for the kind comments and suggestions meant to improve the manuscript.

Comment 1: Please include a rebuttal letter that responds to each point raised by the academic editor and reviewer(s). You should upload this letter as a separate file labelled 'Response to Reviewers'.

Response 1: A rebuttal letter labelled 'Response to Reviewers' has been included per directive of the editor

Comment 2: Please include a marked-up copy of your manuscript that highlights changes made to the original version. You should upload this as a separate file labelled 'Revised Manuscript with Track Changes'.

Response 2: “Track changes” in MS word has been used to highlight the revisions and the file has been saved as 'Revised Manuscript with Track Changes'.

Comment 3: Please an unmarked version of your revised paper without tracked changes. You should upload this as a separate file labelled 'Manuscript'.

Response 3: A clean copy of the Manuscript devoid of highlights and tracked changes has been labelled and saved as 'Manuscript'.

Reviewer 1

General Response: We are grateful to the reviewer for the kind comments and efforts to make the manuscript better. All the corrections have been implemented and can be confirmed easily in the file saved as “Revised manuscript with track changes”. Line numbers (LNs) are also included in the point by point responses to reviewers for efficient tracking of corrections.

Comment 1: Please check the writing format of the scientific name of all microorganisms. There are “Streptococcus” and “Salmonella” which can be abbreviated as “S.”. This would make readers confused. It is suggested to present “S.” for Streptococcus mutans and “Sm”. for Salmonella. To describe Salmonella enterica serovar Typhimurium, it is recommended to state it as “Salmonella var. Typhimurium” or “Salmonella Typhimurium” or “Sm. Typhimurium”. Please correct it in the whole manuscript. Please indicate strain to all microorganisms used in this study. Full name of a microorganism must be present for the first time of the main section and then it should be abbreviated. However, when the authors state it as a subject of the sentence, it is recommended to state its full name. 

Response 1: Thank you for the comment. The writing format for the scientific names have been modified as suggested and the corrections was applied throughout the manuscript.

Comment 2: It is confusing when the authors state NFP and FFP as Miang because either NFP or FFP is a process of Miang fermentation. It is recommended to state it as “NFP Miang” or “FFP Miang”. Please also correct these words throughout the manuscript.

Response 2: “NFP” and “FFP” has been clarified and changed throughout the manuscript to “NFP Miang” or “FFP Miang” (LNs 43, 47, 52, 60, 98, 99, 107, 109, 114, 233, 234, 236, 249, 270,273, 277, 283, 287, 291-293, 342, 352, 371, 376, 380, 392, 396, 412, 435-436, 441-442 and 471) as recommended. 

Comment 3: Comma symbol (,) must be added after a specific word such as respectively, especially, (LNs 70, 152, 161, 286, 292, 303, 381, 413 and 480) etc.

Response 3: Comma symbols have been added as suggested for specific words.

Comment 4: Considering the type of extracts used in this study and their consistency, please reconsider these words: non-neutral, neutral, acidic, and pH 7. If tea leaf extracts (YTL and MTL) and NFP and FFP Miang extracts are generally acidic, it could state them as “non-neutralized extracts”. After the authors have neutralized the acidic extracts, it could state them as “neutralized extracts”. 

Response 4: Thank you for the comment. “Non-neutral extract” and “neutral extract” were replaced with “non-neutralized extracts” and “neutralized extracts” respectively (LNs 131-132, 144, 253, 262, 271, 274, 284, 287, 289, 295, 322 and 477). However, words such as non-neutral pH, and neutral pH were not altered. 

Comment 5: Due to Miang is made from tea leaves, it is relevant to express only young tea leaves or mature tea leaves rather than unfermented young tea leaves and unfermented mature tea leaves, respectively (Line 228 and others in discussion section). 

Response 5: The word “unfermented” in LNs 234, 359, 373, 411 and 440, has been erased due to its redundancy. Thank you for the suggestion. 

Comment 6: Please italicize p of p-value and t of t-value in whole manuscript.

Response 6: All p-values found in LNs 227, 244-245, 249, 278, 280, 293, 303, 308, 316, 329, 335 and 348, have been italicized. 

Comment 7: It is more relevant to replace “PVPP-treated extract’ supernatant” by “PVPP-treated extract”. (Lines 148, 398, 408 …).

Response 7: Thank you for the comment. “PVPP-treated extract’ supernatant” has been changed to “PVPP-treated extract” (LNs 160, 180, 206, 330, 339-340, 433, 438 and 446) throughout the manuscript.

Comment 8: Introduction section, add more information about role of sortase A enzyme in dental plaque formation. This information would at least let readers primarily understand why the authors chose molecular docking to investigate interactions between active compounds in Miang extracts and the sortase A enzyme.

Response 8: Additional information has been included in the Introductory part as recommended: LNs 80-82, “The biofilm formation is mediated through sortase A enzyme which anchors surface proteins on bacterial cell walls to allow host attachment by S. mutans biofilms. Consequently, these events lead to the formation of dental plaque, demineralization of enamel and leaching of teeth components [14,15]”.

Comment 9: Line 109-107 how did the authors express the unit of all extract? Is it mg/mL total polyphenol content or mg/mL total tannin content, or mg extract/mL, etc., Please add more information.

Response 9: The unit of the extract was expressed as mg extract/mL as indicated now in LN 121 as suggested.

 Comment 10: Line 127, is it non-neutralized Miang extracts? Line 139, is it neutralized Miang extracts? Please also consider as above mentioned.

Response 10: Thank you for the comment, the previous recommendation (comment 4) also relates to this and has been judiciously implemented throughout the manuscript.

Comment 11: Line 133, indicate diameter of borer. Line 134, indicate temperature. Line 163, remove “be” Line 172, edit to “Streptococcus mutans” when it was first used as the subject. Line 173, edit to “an optical density at 600 nm (OD600)” Lines 223, 247, and 253 add the strain of S. mutans. 

Response 11: The diameter of the borer: “6mm diameter”, has been indicated (LN 138). Temperature has been indicated on LN 140 “incubated for 24 h at 37 oC”. “be” has been removed from LN 168. Line 177, has been edited to “Streptococcus mutans DMST 18777” due to its use here as the subject. LN 178 has been edited to “to attain an optical density at 600 nm (OD600) of ~1”. The strain of S. mutans has been included in LNs 235, 291 and 308. All the corrections have been implemented and can be confirmed easily in the file saved as “Revised manuscript with track changes”. 

Comment 12: LNs 230 and 239 how did the authors calculate percentages of inhibition? Please clarify.

Response 12: Now LNs 236 and 247, % Contribution of neutralized compounds (Table 1) was expressed as (Zone of inhibition for neutralized extract/ Zone of inhibition for non-neutralized extract) x 100%. Then, contribution of organic acids = 100 % ˗ % Contribution of neutralized compounds.

Comment 13: Table 1, the authors can categorize the extracts into two groups including non-neutralized (acidic) and neutralized extracts. Each group consists of YTL, MTL, NFP, and FFP.

Response 13: Thank you for the comment, the change was duly noted (LNs 238-245) and incorporated in the table.

Comment 14: Line 236-237, different alphabets represent significant differences vs control. The control was just water. If the authors compared the experimental values to the control, extract samples exhibited the clear zone will have significant differences. From the point of view, multiple comparisons were made after ANOVA results showed significant differences (p < 0.05). It is supposed that all pairwise comparisons were used to analyze the data within the same row, not just compared with the control. It is noted that data with positive results are always significant when they are compared with the control. Please consider this comment for Fig 3., Table 2, Fig 5, and Fig 6. 

Response 14: Thank you for the comment, the comment was considered and a modification (“Different alphabets represent significant differences (p<0.05, Turkey’s test).”) was applied accordingly to the affected figures and tables mentioned above (LNs 244, 302, 316-317, 335 and 347).

Comment 15: Lines 245 and 251, it is recommended to revise the figure legend as follows: 

Fig 1. Antibacterial activities of non-neutralized tea leaf extracts and non-neutralized Miang extracts against Bacillus cereus TISTR 747 (A), Escherichia coli ATCC 22595 (B), Salmonella Typhimurium TISTR 292 (C), Streptococcus mutans DMST 18777 (D). Sterile water was used as the control.

Fig 2. Antibacterial activities of neutralized tea leaf extracts and neutralized Miang extracts against Bacillus cereus TISTR 747 (A), Escherichia coli ATCC 22595 (B), Salmonella Typhimurium TISTR 292 (C), Streptococcus mutans DMST 18777 (D). Sterile water was used as the control. 

Response 15: Thank you for the comment. The Figure title has been revised as recommended (LNs 253-269). 

Comment 16: Lines 256-277, it is recommended to initially describe line 257-260 (Antibacterial quantitative assays showed …….at acidic and neutral extracts’ pH respectively). However, the phrase “at acidic and neutral extracts’ pH respectively” should be revised to proper format as previously suggested. For example, Antibacterial quantitative assays showed an overall better MIC and MBC for NFP and FFP Miang extracts compared to the YTL and MTL when they were tested against all the pathogens, respectively.

Response 16: Now LNs 272 – 275, the MIC and MBC was initially defined in Lines 151 and 152. However, it has been modified to “Antibacterial quantitative assays showed an overall better MIC (< 3 log decrease in CFU/mL) and MBC (> 3 log decrease in CFU/mL) for NFP and FFP Miang compared to the young (YTL) and mature tea leaves (MTL) against all the pathogens at non-neutralized and neutralized pH of the extracts respectively”. 

Comment 17: Line 278, it is recommended to revise the figure legend as follows: 

Fig 3. Antibacterial activities of non-neutralized and neutralized extracts of Miang and tea leaves against Bacillus cereus TISTR 747 (A), Escherichia coli ATCC 22595 (B), Salmonella Typhimurium TISTR 292 (C), Streptococcus mutans DMST 18777 (D). 

Response 17: The figure title has been revised as recommended (LN 295).

Comment 18: Line 282, Different asterisk (*) represent significant differences vs control. What was the control of this experiment? Is it the initial cell concentration or just sterile water? This phrase could be modified to “Different asterisk * and ** represent significant differences of MIC and MBC, respectively (p < 0.05)”.

Response 18: The initial cell concentration and sterile water were both incorporated in the experimental design. However, “Different asterisk * and ** represent significant differences of MIC and MBC, respectively (p<0.05)” better captures the multiple comparison from Turkey’s post hoc test. Thank you very much for this comment (LN 302).

Comment 19: Line 289, please consider modifying description of table 2 as follows:

Table 2. The inhibition zone of tannin-free extracts of Miang and tea leaves against Streptococcus mutans DMST 18777. Avoid extension of figure description. 

Response 19: Thank you for the comment. The simpler description has been implemented (LN 310) as suggested.

Comment 20: Line 293 the comment is similar to that is given for Line 236-237. Line 296, it is unnecessary to add Fig 4 because the authors have already displayed the results in Table 2.

Response 20: “Different alphabets…” indicating the multiple comparisons from Turkey’s test have been modified throughout the manuscript as recommended. However, we feel some readers resonate more with figures so we showed the fig 2 in addition to the table (which shows the statistical difference) in a bid to accelerate readers’ understanding of the result.

Comment 21: Lines 309-314 it is necessary to compare effect of tannin-free extracts and their original extract (in the presence of tannin)

Response 21: Thank you for the comment, the effect of tannin-free extracts and their original extracts were compared in table 2 where independent sample T-test was used to explain the difference. Against the S. mutans DMST 18777 biofilms, the point of this publication was to explore the potential of tannin free extracts in the growth inhibition of the biofilms. The justification for this was mentioned in the introduction (Lines 90 – 97).

Comment 22: Lines 384-386 Streptococcus mutans has been reported to recruit sucrose-dependent mechanisms where glucosyltransferase (GTF) reacts with sucrose and convert into dextran polymer or glucan that can be further bound with glucan-binding proteins (GBP) which leads to the formation of dental plaque. Please consider modifying the sentence.

Response 22: Lines 394 – 397), the sentence has been modified to “Streptococcus mutans has been reported to recruit sucrose-dependent mechanisms where glucosyltransferase (GTF) mediates a reaction with glucan-binding proteins (GBP) to cause the formation of dental plaques [49–51]. Interestingly, tannins have been reported to inhibit GTF-mediated synthesis of insoluble glucans thus suppressed S. mutans proliferation”.

Comment 23: Lines 411-425 please shorten principal discussion about role of sortase A in plaque formation but discuss more about finding of this study. How could active compounds from PVPP-treated Miang extracts inhibit sortase A? Are there any literatures reported active compounds that have potential to inhibit the enzyme? 

Response 23: The discussion was made to be as concise as possible (Lines 448 – 465). And a Paper which was recently published (December, 2022) was cited which mentioned related active compounds that have found use in maintaining oral health. Line 466 – 469, “Medicinal plants rich in flavonoids (including kempferol, apigenin and quercetin) and tannins (such as ellagitannins and proanthocyanidins) have been shown to contribute to the maintenance of oral health and are therefore applied as additives or main ingredients in mouthwashes or toothpastes [59]”. 

Comment 24: Line 438-440 the sentence in the conclusion is not relevant to what the author proposes in introduction section (Lines 51-53, and 90-93).

Response 24: The sentence has been modified to “Overall, the result showed the potential of these extracts to be enriched, optimized and possibly find future applications as a mainstay in combating bacterial pathogens and as potential ingredients in mouthwashes” (484 – 486).

Reviewer 2

General Comment: It was advantageous to discover a plant extract that inhibited pathogen growth. Several concerns, however, needed to be elucidated in order to have a better knowledge of Miang extracts' antibacterial activity.

General Response: Thank you very much for the comment. Your concerns will be given due attention and addressed appropriately. All corrections by the authors are made using Tracked changes and saved as “Revised document with Track changes”. Line numbers (LNs) are also included in the point by point responses to reviewers for effic

---

## [Decision Letter · Decision Letter 1]

20 Mar 2024

PONE-D-23-35234R1Antibacterial activities of Miang extracts against selected pathogens and the potential of the tannin-free extracts in the growth inhibition of Streptococcus mutansPLOS ONE

Dear Dr. Khanongnuch,

Thank you for submitting your manuscript to PLOS ONE. After careful consideration, we feel that it has merit but does not fully meet PLOS ONE’s publication criteria as it currently stands. Therefore, we invite you to submit a revised version of the manuscript that addresses the points raised during the review process.

**ACADEMIC EDITOR:**

Dear author, I received the opinion on your manuscript. Review reviewers' recommendations point by point

We look forward to receiving your revised manuscript.

Kind regards,

Mozaniel Santana de Oliveira, Ph.D

Academic Editor

PLOS ONE

Journal Requirements:

Reviewers' comments:

Reviewer's Responses to Questions

**Comments to the Author**

1. If the authors have adequately addressed your comments raised in a previous round of review and you feel that this manuscript is now acceptable for publication, you may indicate that here to bypass the “Comments to the Author” section, enter your conflict of interest statement in the “Confidential to Editor” section, and submit your "Accept" recommendation.

Reviewer #1: All comments have been addressed

Reviewer #2: (No Response)

Reviewer #3: All comments have been addressed

2. Is the manuscript technically sound, and do the data support the conclusions?

Reviewer #1: Yes

Reviewer #2: Partly

Reviewer #3: Yes

3. Has the statistical analysis been performed appropriately and rigorously? 

Reviewer #1: Yes

Reviewer #2: N/A

Reviewer #3: Yes

4. Have the authors made all data underlying the findings in their manuscript fully available?

Reviewer #1: No

Reviewer #2: Yes

Reviewer #3: Yes

5. Is the manuscript presented in an intelligible fashion and written in standard English?

Reviewer #1: Yes

Reviewer #2: Yes

Reviewer #3: Yes

6. Review Comments to the Author

Reviewer #1: Dear Authors,

Thank you for your revised manuscript. All of my comments and suggestions have been almost completed. It would be great if the authors could include response 12 in the manuscript.

Reviewer #2: Although the authors responded to the comments, some points needed to be elucidated.

The authors should determine which active ingredients in Miang extract prevent antibacterial growth.

To describe the study's findings, the authors should include the concentration of Miang extract that was observed to limit bacterial growth.

Reviewer #3: All the comments that I raised have been addressed adequately. I comend the authors for the efforts put into addressing all the comments posed by the reviewers.

7. PLOS authors have the option to publish the peer review history of their article (what does this mean?). If published, this will include your full peer review and any attached files.

Reviewer #1: No

Reviewer #2: No

Reviewer #3: No

---

## [Author Response · Author response to Decision Letter 1]

23 Mar 2024

Reviewer 1

General Response: We are grateful to the reviewer for the kind comments and efforts to make the manuscript better. All the corrections have been implemented and can be confirmed easily in the file saved as “Revised manuscript with track changes”. Line numbers (LNs) are also included in the point by point responses to reviewers for efficient tracking of corrections. Responses to R1 are highlighted in yellow in the “Revised manuscript with track changes”.

Comment: Dear Authors, thank you for your revised manuscript. All of my comments and suggestions have been almost completed. It would be great if the authors could include response 12 in the manuscript.

Response: Thank you very much for the comment. Response 12 from the major revision responded the question on how percentage inhibition was calculated. The response has been included in materials and methods section on LNs 140 – 142, “Inhibition by neutralized extract and organic acids is calculated according to the following equation: inhibition by neutralized extract (%) = (zone of inhibition for neutralized extract/zone of inhibition for non-neutralized extract) × 100, and inhibition by organic acids (%) = 100 ˗ inhibition by neutralized extract”.

Reviewer 2

General Response: Thank you very much for the comment. All corrections by the authors are made using Tracked changes and saved as “Revised document with Track changes”. Line numbers (LNs) are also included in the point by point responses to reviewers for efficient tracking of corrections. Responses to R2 are highlighted in green in the “Revised manuscript with track changes”.

Comment: Although the authors responded to the comments, some points needed to be elucidated. The authors should determine which active ingredients in Miang extract prevent antibacterial growth. To describe the study's findings, the authors should include the concentration of Miang extract that was observed to limit bacterial growth.

Response 1: We already established previously (Reference 7, LN 502) that tannins/proanthocyanidins are major bioactive compounds in Miang and play important roles as antioxidant and antimicrobial agents. In the context of this study, we suggested that ellagic acid, flavonoid aglycones, flavonoid glycosides and organic acid derivatives in tannin-free extracts (See supplementary information) as probably responsible for the observed antibacterial activities against S. mutans DMST 18777. LN 441 – 444, “Our results suggested ellagic acid, flavonoid aglycones, flavonoid glycosides and organic acid derivatives might be responsible for the antimicrobial activities of NFP and FFP Miang (Table 3, Fig 7C) from the observed docking binding energies”.

In addition, the concentrations of Miang extracts that inhibited bacterial growth were represented as minimum inhibitory concentration (MIC) and minimum bactericidal concentration (MBC) as seen in LNs 264 – 287 (Result section) and illustrated in Fig 3. Future studies will try evaluating the potential activities of these suggested compounds against the bacterial pathogen.

---

## [Decision Letter · Decision Letter 2]

9 Apr 2024

Antibacterial activities of Miang extracts against selected pathogens and the potential of the tannin-free extracts in the growth inhibition of Streptococcus mutans

PONE-D-23-35234R2

Dear Dr. Khanongnuch,

We’re pleased to inform you that your manuscript has been judged scientifically suitable for publication and will be formally accepted for publication once it meets all outstanding technical requirements.

Kind regards,

Mozaniel Santana de Oliveira, Ph.D

Academic Editor

PLOS ONE

Additional Editor Comments (optional):

Reviewers' comments:

Reviewer's Responses to Questions

**Comments to the Author**

1. If the authors have adequately addressed your comments raised in a previous round of review and you feel that this manuscript is now acceptable for publication, you may indicate that here to bypass the “Comments to the Author” section, enter your conflict of interest statement in the “Confidential to Editor” section, and submit your "Accept" recommendation.

Reviewer #2: All comments have been addressed

2. Is the manuscript technically sound, and do the data support the conclusions?

Reviewer #2: Yes

3. Has the statistical analysis been performed appropriately and rigorously? 

Reviewer #2: Yes

4. Have the authors made all data underlying the findings in their manuscript fully available?

Reviewer #2: Yes

5. Is the manuscript presented in an intelligible fashion and written in standard English?

Reviewer #2: Yes

6. Review Comments to the Author

Reviewer #2: The authors had responded to all comments. Please remove value 3.76 from Table 3, as it did not reflect any parameter.

7. PLOS authors have the option to publish the peer review history of their article (what does this mean?). If published, this will include your full peer review and any attached files.

Reviewer #2: No

---

## [Editor Report · Acceptance letter]

26 Apr 2024

PONE-D-23-35234R2 

PLOS ONE

Dear Dr. Khanongnuch, 

I'm pleased to inform you that your manuscript has been deemed suitable for publication in PLOS ONE. Congratulations! Your manuscript is now being handed over to our production team.

Kind regards, 

on behalf of

Dr. Mozaniel Santana de Oliveira 

Academic Editor

PLOS ONE